# Microbiota Metabolite Profiles and Dietary Intake in Older Individuals with Insomnia of Short vs. Normal Sleep Duration

**DOI:** 10.3390/biom14040419

**Published:** 2024-03-30

**Authors:** Carmel Even, Faiga Magzal, Tamar Shochat, Iris Haimov, Maayan Agmon, Snait Tamir

**Affiliations:** 1Nutritional Science Department, Tel Hai College, Upper Galilee, Kiryat Shmona 1220800, Israelsnait@telhai.ac.il (S.T.); 2Laboratory of Human Health and Nutrition Sciences, MIGAL-Galilee Research Institute, Kiryat Shmona 11016, Israel; 3The Cheryl Spencer Department of Nursing, University of Haifa, Haifa 3103301, Israel; tshochat@univ.haifa.ac.il (T.S.);; 4Department of Psychology and the Center for Psychobiological Research, The Max Stern Yezreel Valley College, Affula 19300, Israel; i_haimov@yvc.ac.il

**Keywords:** amino acids, deoxycholic acid, diet, insomnia, microbiota metabolites, sleep duration

## Abstract

Recent evidence suggests that the gut microbiota plays a role in insomnia pathogenesis. This study compared the dietary habits and microbiota metabolites of older adults with insomnia of short vs. normal sleep duration (ISSD and INSD, respectively). Data collection included sleep assessment through actigraphy, dietary analysis using the Food Frequency Questionnaire, and metabolomic profiling of stool samples. The results show that ISSD individuals had higher body mass index and a greater prevalence of hypertension. Significant dietary differences were observed, with the normal sleep group consuming more kilocalories per day and specific aromatic amino acids (AAAs) phenylalanine and tyrosine and branch-chain amino acid (BCAA) valine per protein content than the short sleep group. Moreover, metabolomic analysis identified elevated levels of the eight microbiota metabolites, benzophenone, pyrogallol, 5-aminopental, butyl acrylate, kojic acid, deoxycholic acid (DCA), trans-anethole, and 5-carboxyvanillic acid, in the short compared to the normal sleep group. The study contributes to the understanding of the potential role of dietary and microbial factors in insomnia, particularly in the context of sleep duration, and opens avenues for targeted dietary interventions and gut microbiota modulation as potential therapeutic approaches for treating insomnia.

## 1. Introduction

Insomnia is a common chronic health condition and the most common sleep disorder, with symptoms affecting approximately 50% of adults over the age of 65 [1]. Insomnia, as defined by the *Diagnostic and Statistical Manual of Mental Disorders, 5th edition* (DSM-5) [2], is a sleep–wake disorder characterized by dissatisfaction with sleep quality or quantity. It involves difficulty initiating or maintaining sleep, early morning awakenings, nonrestorative sleep and significant distress or impairment in various areas of life [2,3,4]. The symptoms must occur for at least three nights per week for a duration of at least three months, and should not be better explained by another sleep, mental, or medical disorder [1]. Insomnia in the older population is linked to a range of disease risks such as diabetes, metabolic syndrome, cognitive impairment, and depression, imposing high costs on healthcare systems and society [5]. 

Evidence supports a distinction between two phenotypes of insomnia defined based on a short vs. normal sleep duration. Insomnia with short sleep duration (ISSD), typically defined as a total sleep time of ≤6 h [6,7], is the most biologically severe phenotype and is associated with physiological hyperarousal and increased risk of cardiometabolic morbidity and mortality [6,8,9]. In contrast, insomnia of normal sleep duration (INSD) (typically defined as a total sleep time of >6 h) is associated with an overall lower risk of cardiometabolic outcomes compared to ISSD [6,7]. Moreover, INSD individuals are characterized by cognitive–emotional and cortical arousal. Because their insomnia is more closely linked to cognitive and emotional factors rather than physiological ones, these individuals are likely to respond better to Cognitive Behavioral Therapy for Insomnia (CBT-I) [10].

Growing evidence shows that the microbiota gut–brain axis contributes to the regulation of sleep behavior both directly and indirectly and may play a critical role in the etiology and pathogenesis of sleep disorders [11]. It has been shown that in older people with insomnia, differences in the composition of the gut microbiota and an abundance of particular genera, such as *Lactobacillus crispatus* and *Streptococcus*, were found to be associated with poor sleep and poor cognitive function [12]. Five metabolic pathways, including those for glycerophospholipid metabolism, glutathione metabolism, nitrogen metabolism, aspartate, glutamate, alanine metabolism, and aminoacyl-tRNA production, may be involved in the link between the gut microbiota and insomnia [13].

The gut microbiota communicates with the brain through the microbiota gut–brain axis (MGBA) [14]. The MGBA comprises neuronal, immune, metabolic, and endocrine pathways [11,15]. Neurotransmitters and metabolites produced by gut microbes, such as gamma-aminobutyric acid (GABA), dopamine, and serotonin (5-HT), can affect neurons of the enteric nervous system (ENS) and interact with afferent pathways of the vagal nerve, affecting the neural circuits involved in sleep–wake regulation [16]. Similarly, gut-derived metabolites can be transmitted to the brain through blood circulation and afferent vagal pathways to affect sleep [17]. Indeed, alterations in the gut microbiota composition have been correlated with specific serum metabolites [18,19], sleep quality and cognitive performance [20] in patients with insomnia.

Gut microbiota metabolites originate from the breakdown of diet nutrients, mainly diet fibers. Proteins, on the other hand, are mostly digested and absorbed in the small intestine. Proteins that escape digestion reach the colonic lumen, where they serve as fermentable substrates for the gut microbiota and undergo proteolysis into amino acids. Colonic amino acids may be further metabolized by the gut microbiota, as exemplified by aromatic amino acids (AAAs) [21]. AAA catabolism by the gut microbiome yields numerous metabolites that can regulate immune, metabolic, and neuronal responses at local and distant sites. Phenylalanine (Phe) is a metabolic precursor of tyrosine (Tyr) via phenylalanine hydroxylase in the liver, which further metabolizes to catecholamines (dopamine, norepinephrine and epinephrine) [22].

Other key metabolites produced by the gut microbiota include secondary bile acids (BAs). The primary BAs pool in humans consists of cholic acid, chenodeoxycholic acid, and subsequent C24 taurine- or glycine-bound derivatives, synthesized in the liver from cholesterol [23]. Primary BAs are heavily modified in the lower gastrointestinal tract to produce a broad range of secondary BAs [23]. Abnormally high levels of the microbially modified secondary BA deoxycholic acid (DCA) are associated with gut dysbiosis and disease [24,25].

Although research on the involvement of gut microbiota metabolites is mounting, the possible involvement of gut metabolites in sleep regulation and insomnia remains largely unknown. Previous studies have shown that individuals with insomnia exhibit distinct metabolic profiles compared to matched controls, including alterations in branched-chain amino acid (BCAA) metabolism and lactate peak timing and amplitudes [1]. It is critical to note that while these studies offer valuable insights into the relationship between sleep and metabolism, they focus predominantly on metabolomic analysis of blood samples rather than fecal samples. This study compared the dietary intake and fecal metabolite profiles of individuals with short vs. normal sleep duration insomnia. As ISSD is essentially a more physiological phenotype of the disorder, we hypothesized that gut metabolites associated with dysbiosis would be upregulated in short sleepers. In addition, we expected lower levels of vital dietary nutrients known to influence sleep in older adults with ISSD.

## 2. Materials and Methods

The study protocol was approved by the institutional review board (IRB) of the Faculty of Social Welfare and Health Sciences at the University of Haifa (approval number 026/17). All participants involved in the study provided informed consent.

### 2.1. Study Cohort

In this study, a focused cohort of 25 participants was derived from a broader preliminary study involving 59 participants [26]. Participants were categorized into two different types of insomnia, INSD and ISSD, using the SPSS K-means clustering analysis, based on two key indicators: sleep efficiency and sleep duration, as measured by actigraphy [26]. This subset was meticulously chosen based on stringent inclusion criteria to ensure a homogeneous sample that accurately reflects distinct insomnia sleep duration phenotypes, short vs. normal. The inclusion criteria for insomnia were sleep onset latency (SOL) or wake after sleep onset (WASO) of >30 min; and less than 85% sleep efficiency (SE, percentage of total sleep time after initial sleep onset out of total time in bed) for at least three out of seven nights each week [26]. The exclusion criteria were chronic pain; substantial and unstable medical, neurological, or psychiatric illness; any significant visual or hearing impairments; psychiatric medication use; alcohol or drug use; sleep apnea syndrome (SAS); and periodic limb movement disorder during sleep (PLMD), based on a self-report. Participants were tested on the Mini Mental State Examination (MMSE) [27] using a cutoff >26 to exclude participants with cognitive impairment.

### 2.2. Sleep Assessment

Sleep was assessed through a clinical interview conducted by telephone by a trained interviewer. Participants were asked about medical condition(s), medication(s), or other substance use, and responded to questions about specific nighttime sleep problems such as frequency and duration of insomnia and difficulties falling asleep or staying asleep. In addition, participants were instructed to wear an Actiwatch, an activity wrist monitor, for a period of two weeks, and total sleep time (TST), SOL, SE, and WASO were collected.

### 2.3. Food Frequency Questionnaire

The dietary assessment was performed using the 127-item Food Frequency Questionnaire (FFQ). The development and validation process of this questionnaire is described in detail elsewhere [28]. Briefly, the FFQ includes 127 food items with nine frequency options ranging from “never or less than once monthly” to “six or more times daily”. The questionnaire is semi-quantitative, and a standard portion size is described for each food item. The portion-size estimates are based on information from the Israeli Ministry of Health. Participants are asked to report their average frequency of consumption during the past year.

### 2.4. Untargeted Metabolomics

Recruited individuals were asked to provide a morning stool sample, which was stored at −20 °C until analysis. A modified liquid chromatography–mass spectrometry (LCMS) analysis protocol [29] was used for stool sample preparation and analysis. In brief, samples were homogenized, frozen, dried, and then pulverized. The samples were then centrifuged, and the upper liquid was transferred for filtration and subsequent LC-MS analysis.

The samples were injected Into an UHPLC connected to a photodiode array detector (Dionex Ultimate 3000, (Thermo Fisher Scientific, San Diego, CA, USA) with a reverse-phase column (ZORBAX Eclipse Plus C18, 3.0 × 100 mm, 1.8 µm, Agilent Technologies, Santa Clara, CA, USA). MS/MS analysis was performed with a heated electrospray ionization source connected to a Q Exactive™ Plus Hybrid Quadrupole-Orbitrap™ Mass Spectrometer (Thermo Fisher Scientific, San Diego, CA, USA). The gradient was initiated with 2% B, which was increased to 30% B over a period of 4 min, and then increased to 40% B over 1 min before being kept isocratic at 40% B for another 3 min. Then, the gradient increased to 50% over 6 min, to 55% over another 4 min, and to 95% over 5 min, and was kept isocratic for 7 min. Finally, phase B was returned to 2% over 3 min and the column was allowed to equilibrate at 2% B for 3 min before the next injection. The flow rate was 0.4 mL/min. Blank (methanol) and QC samples were injected at the start of the sequence, after every 10 samples, and at the end of the sequence.

LC–MS/MS analysis was performed with a Heated Electrospray ionization (HESI-II) source connected to a Q Exactive™ Plus Hybrid Quadrupole-Orbitrap™ Mass Spectrometer, Thermo Scientific™, Dreieich, Hessen Germany. The ESI capillary voltage was set to 3500 V, the capillary temperature to 300 °C, gas temperature to 350 °C, and the gas flow to 10 mL/min. The mass spectra (*m*/*z* 100–1500) were acquired using both positive and negative ion modes. Data-dependent MS2 analysis was generated for the QC samples and used for compound identification. Downstream analysis and data processing were performed with the Thermo Scientific™ Compound Discoverer™ program, version 3.1.0.305 (mass tolerance ≤ 5 ppm; intensity tolerance ≤ 30%; S/N threshold = 3; minimum peak intensity = 1,000,000; RT tolerance ≤ 0.2 min). Databases used for identification were Chemspider [30], MzCloud [31] and KEGG [32].

### 2.5. Statistical Analysis

Statistical tests were employed to examine differences in the prevalence of metabolites, sleep measures, demographic characteristics, quality of life and dietary consumption between the short and normal groups. Data were processed and analyzed using various statistical tools including independent-samples *t*-tests, nonmetric dimensional scaling (NMDS), ANOSIM, MANOVA, and Mann–Whitney tests, using software such as Compound Discoverer (version 3.1) (Thermo Fisher Scientific, San Diego, CA, USA), R software (version 3.6.2) https://www.r-project.org/, SPSS software (version 20.0) (Add reference present in comment), and GraphPad Prism 8 software (version 8.3.1) https://www.graphpad.com.

## 3. Results

### 3.1. Demographic and Sleep Characteristics

The demographic and sleep characteristics of the study cohort are shown in Table 1. The average age of the entire study population was 74.9 ± 6.9 years, and most of the participants were female (91.7%). No significant differences were detected between the normal and short groups in terms of age, gender distribution, educational background, marital status or living status (alone or accompanied). The short sleepers, however, demonstrated a significantly higher body mass index (BMI) (33.91 ± 7.77 kg/m^2^) than normal sleepers (24.79 ± 4.18 kg/m^2^; *p* = 0.0011). In addition, short sleepers had a significantly higher prevalence of hypertension (77.8%) compared to the normal sleep group (20.0%) (Chi-squared *p* = 0.0053). The prevalence of diabetes and usage of medication for sleep or depression, as well as usage of anticholinergic medications, were similar across groups (Table 1).

Normal sleepers tended to be more physically active (100%) compared to short sleepers (77.8%; *p* = 0.0573). The television and computer screen time were similar between groups (Table 1).

TST and ST were the two key parameters used to define INSD and ISSD. No significant differences were noted for other sleep parameters, such as SOL, SE, ET, and WASO.

### 3.2. Significant Differences in Nutritional Intake between INSD and ISSD Participants

The mean participant nutritional intake of the two study groups is summarized in Table 2 and Appendix A. Out of the 25 participants included in this study, 19 completed the FFQ questionnaire (*n* = 11 and 8, normal and short group, respectively) and were included in the analysis. The participants in the normal group consumed significantly more kilocalories throughout the day than those in the short group (*p* = 0.023), while the relative consumption of proteins, carbohydrates, and fats (%/total kcal) was not significantly different between the groups (*p* > 0.05). Since ISSD had a significantly lower energy intake in comparison with the ISND group, lower levels of all macro and micronutrients were expected to be present in the ISSD group, creating bias. Therefore, we calculated the % of each dietary compound within its category (e.g., % amino acids/total protein), and performed a statistical analysis between both groups. The results show that, except for glutamic acid, proline, serine, hydroxyproline, leucine, isoleucine and tryptophan, the % of all amino acids/total protein was significantly higher in the normal group compared to the short group (*p* < 0.05). No differences in fatty acid, mineral, and vitamin percentages were detected between groups (Appendix A).

### 3.3. LCMS Analysis

#### 3.3.1. Distinct Metabolite Compounds Associated with INSD and ISSD Participants

We conducted untargeted LCMS to analyze fecal samples collected from INSD and ISSD participants (*n* = 15 and *n* = 10, respectively). We found 19,502 compounds, of which 1451 were identified as different metabolites (with names and without isomers) (Appendix A). Among the 1451 metabolites identified, 82 were significantly upregulated, and 3 were significantly downregulated in the ISSD group (Figure 1).

#### 3.3.2. Similar Metabolite Profile between INSD and ISSD Participants

We performed a nonmetric multidimensional scaling (NMDS) of Bray–Curtis dissimilarity to test for metabolite dissimilarity between INSD and ISSD participants (Figure 2). The samples from the normal and short groups were found to be almost uniformly distributed across the center of the plot, suggesting that the metabolite profiles of these two groups were not markedly different. An ANOSIM statistical test confirmed the absence of a significant difference in the metabolite profile of the two study groups (*p* > 0.05).

#### 3.3.3. Upregulated Metabolites Linked to Bacterial Metabolic Pathways in ISSD Participants

From the 82 metabolites found to be significantly upregulated in the ISSD group in comparison to the INSD group, eight (benzophenone, pyrogallol, 5-aminopental, butyl acrylate, kojic acid, deoxycholic acid (DCA), trans-anethole, and 5-carboxyvanillic acid) were linked to bacterial metabolic pathways (Table 3).

## 4. Discussion

This study aimed to gain further insights into the possible contribution of microbiome metabolites to short or normal sleep duration phenotypes of insomnia in older adults. Significant differences were found in the BMI, blood pressure, amino acid consumption and gut microbiome metabolite profiles of individuals with short vs. normal sleep-duration insomnia. Our main findings are (i) older adults with ISSD consumed lower levels of amino acids such as phenylalanine, tyrosine, and valine, which are known to play a role in sleep regulation and (ii) DCA, a gut metabolite associated with dysbiosis and disease, was upregulated in ISSD older adults. Overall, these findings support our study hypothesis.

Short sleepers had, on average, higher BMI and hypertension compared to normal sleepers. Several studies have suggested that short sleep duration is a risk factor for hypertension and metabolic diseases [33,34,35,36,37,38,39]. For instance, Gangwisch et al. [34] reported a positive association between short sleep duration and hypertension in adults, even after adjusting for BMI and diabetes history. A study by Wang et al. indicated that both short and long sleep durations are associated with higher risks of hypertension [35]. For patients with insomnia, those with objectively short sleep duration are particularly susceptible to developing hypertension [36] and cardiovascular disease, possibly due to their state of hyperarousal, indicated by elevated hypothalamic–pituitary–adrenal axis (HPA) and sympathetic activity [7]. Short sleep duration is also associated with the occurrence of overweight and obesity, with sleep duration of less than seven hours linked to a nearly two-fold increased rate of overweight and obesity compared to a 7–9 h sleep duration [38]. Furthermore, studies have shown that individuals with shorter sleep duration tend to have higher BMI [40].

Diet, particularly the intake of micronutrients, such as amino acids, is a crucial factor influencing sleep quality and can contribute to insomnia [41]. In the present study, although the protein percentage intake was the same between groups, short sleepers consumed relatively lower levels of the aromatic amino acids (AAAs) phenylalanine and tyrosine compared to the normal group. These findings are in line with a recent study which showed a significantly positive association between sleep duration and the intake of AAAs in adults with normal BMI [42]. Furthermore, significantly less valine, a branch-chain amino acid (BCAA), was consumed by the participants with a short sleep duration. BCAAs are involved in the de novo synthesis of glutamate and GABA, two neurotransmitters known to influence sleep [43,44]. Indeed, there is some evidence in patients with insomnia suggesting a correlation between reduced GABA levels and shorter sleep duration [45].

To investigate the role of microbiota-derived metabolites in insomnia, this study examined metabolites from the “microbial metabolism in diverse environments” pathway, specifically benzophenone, pyrogallol, 5-aminopental, butyl acrylate, kojic acid, DCA, trans-anethole, and 5-carboxyvanillic acid. From these metabolites, only secondary bile acids, such as DCA, have been previously reported to be related to sleep. DCA, in the current analysis, was detected in higher levels in short sleepers. DCA is generated as a result of biotransformation (7α/β-dehydroxylation) of the primary bile acid, cholic acid, by specific bacteria in the intestinal microbiota, such as species belonging to the *Clostridium* genus (e.g., *C. scindens*, *C. hiranonis*, *C. hylemonae* and *C. sordelli*) [46]. Unlike rodents, the human liver is incapable of 7α-hydroxylating secondary bile acids returning to the liver via the portal vein, and, thus, secondary bile acids accumulate to high levels in the bile of some humans [47]. The current findings are in line with a recent study which integrated multi-omics data from two human cohorts and which found an association between insomnia and higher levels of secondary bile acids [48], suggesting that insomnia might have a significant impact on the gut microbiota–bile acid axis.

Higher levels of DCA, shown here in short sleepers, have been previously shown to cause oxidative stress in the intestinal epithelial cells, leading to increased secretion of inflammatory cytokines, such as IL-1β, IL-6, and TNF-α, in colonocytes and causing damage to the mucosal layer of the colon [25]. Furthermore, mice fed with DCA developed inflammation in the colon, accompanied by dysbiosis of the intestinal microbiota [49]. Given that insomnia of short sleep duration is also positively associated with inflammation and dysbiosis, DCA could be considered a potential biomarker for this disease phenotype.

The strengths of the study included its comprehensive approach, combining various methods such as sleep assessment, dietary analysis, and advanced metabolomics, to gain a detailed understanding of the interplay between sleep patterns, diet and metabolism. The study was limited by the small sample size, which may impact the generalizability of the findings, the self-reported dietary data which could introduce bias, and the focus on older adults which limits generalization to younger age groups.

## 5. Conclusions

Overall, this study found that older adults with short sleep-duration insomnia had higher BMI and blood pressure compared with the normal sleep-duration group. Moreover, short sleepers consumed fewer amino acids and presented higher fecal levels of DCA, a secondary bile acid produced by microbiota fermentation. Further investigation will be needed to elucidate the mechanisms connecting sleep, diet, and microbiota metabolites and the role of DCA as a biomarker in insomnia.

## Figures and Tables

**Figure 1 biomolecules-14-00419-f001:**
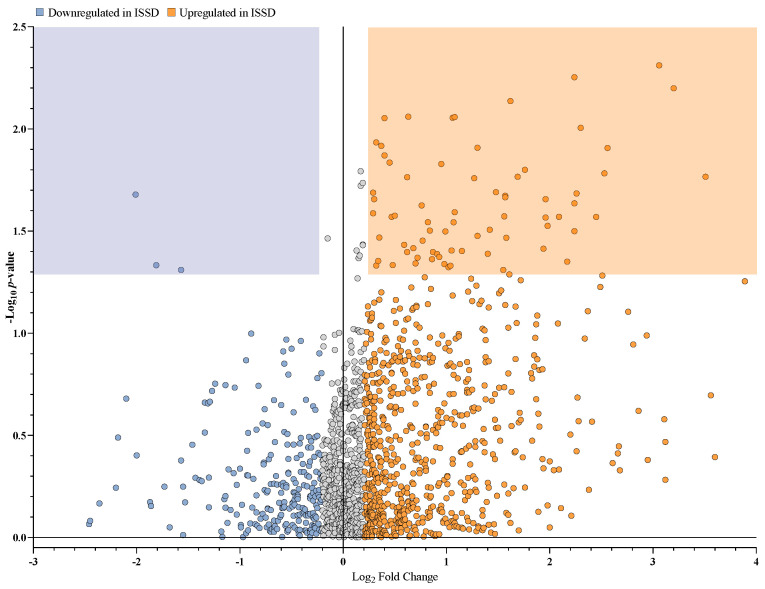
Upregulated and downregulated metabolites in fecal samples of ISSD participants. Each data point in the volcano plot represents a single metabolite, with the *x*-axis indicating the average relative change or fold change (in log2 scale) between downregulated (blue) and upregulated (orange) metabolites in ISSD participants. The *y*-axis represents the *p*-value (in −log10 scale) for the relative change for each metabolite. A Log2 fold change of less than −0.2 and a *p*-value of less than 0.05, are commonly accepted for statistical significance. The orange and blue areas represent metabolites that were significantly upregulated and downregulated in the ISSD group, respectively. Grey data points represent metabolites that were not significantly different. ISSD = Insomnia of short sleep duration.

**Figure 2 biomolecules-14-00419-f002:**
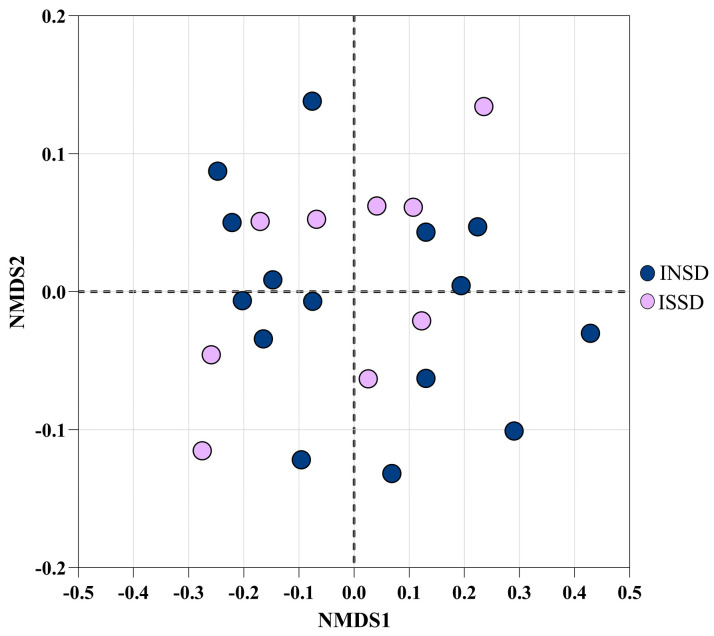
Nonmetric multidimensional scaling (NMDS) of Bray–Curtis dissimilarity (Î^2^ diversity) in INSD and ISSD participants. Dark blue and purple points represent INSD and ISSD participants, respectively.

**Table 1 biomolecules-14-00419-t001:** Characteristics of the study population.

Parameter	Study Population (*n* = 25)	INSD (*n* = 15)	ISSD (*n* = 10)	*p*
**Age** (years)	74.88 ± 6.94	74.2 ± 8.42	75.90 ± 4.01	0.560 ^1^
**Gender** (%)				
Female	91.7	86.7	100	0.229 ^3^
Male	8.3	13.3	0	
**Education** (years)	17.0 ± 2.0	17.0 ± 2.0	17.0 ± 3.0	0.892 ^2^
**Marital status** (%)				
Married	64.0	60.0	66.7	0.744 ^3^
Other	36.0	40.0	33.3	
**Living status** (%)				
Lives with roommates	64.0	60.0	66.7	0.744 ^3^
Lives alone	36.0	40.0	33.3	
**BMI** (kg/m^2^)	28.59 ± 7.39	24.79 ± 4.18	33.91 ± 7.77	**0.001 ^1^**
**Metabolic syndromes (%)**				
Diabetes	20.8	13.3	33.3	0.243 ^3^
Hypertension	41.7	20	77.8	**0.005 ^3^**
Heart disease	8.3	6.7	11.1	0.703 ^3^
**Medication use** (%)				
Sleep medications	20	23.1	14.3	0.639 ^3^
Depression medications	20	30.8	0	0.101 ^3^
Anticholinergic medications	15	7.7	28.6	0.375 ^3^
**Physical activity** (%)	91.7	100	77.8	0.057 ^3^
**Screen time during free time** (min)				
Television	134.38 ± 78.81	130.00 ± 76.63	141.67 ± 86.53	0.734 ^1^
Computer	87.08 ± 61.68	89.33 ± 57.51	83.33 ± 71.59	0.726 ^2^
**Sleep measurements**				
TST (min)	403.84 ± 67.91	446.34 ± 41.85	340.08 ± 45.24	**<0.001 ^1^**
SOL (min)	17.81 ± 15.03	15.94 ±12.66	20.61 ± 18.41	0.723 ^2^
SE (%)	82.16 ± 7.71	84.02 ± 4.04	79.37 ± 10.89	0.196 ^2^
ST (time of the day in decimal number)	24.36 ± 1.33	23.79 ± 0.94	25.22 ± 1.40	**0.005 ^1^**
ET (time of the day in decimal number)	7.00 ± 1.09	7.23 ± 0.94	6.67 ± 1.26	0.217 ^1^
WASO (min)	49.31 ± 16.61	51.26 ± 14.22	46.39 ± 20.13	0.484 ^1^

The values are presented in the table as mean ± standard deviation (mean ± SD), except the values of the variables gender, marital status, accommodation, metabolic syndromes, medication use, and physical activity, which are shown in frequency (%). *p* represents the level of significance of the differences between the normal group and the short group, as determined by ^1^ independent-samples *t*-test, ^2^ Mann–Whitney, ^3^ Chi-squared. TST—total sleep time; SOL—sleep onset latency; SE—sleep efficiency; ST—start time; ET—end time; WASO—wake after sleep onset; INSD = insomnia of normal sleep duration; ISSD = insomnia of short sleep duration. ***p*** < 0.05 are shown in **bold**.

**Table 2 biomolecules-14-00419-t002:** Nutritional intake of the participants with INSD vs. ISSD, as measured by the Food Frequency Questionnaire (FFQ).

Nutrient	INSD (*n* = 11)	ISSD (*n* = 8)	*p*
**Total Energy (kcal/day)**	2423.8 ± 693.1	1775.6 ± 434.9	0.023
**Macronutrients (%/kcal)**			
*Proteins*	17.47 ± 2.68	17.57 ± 2.36	0.932
*Carbohydrates*	48.15 ± 7.34	48.07 ± 6.51	0.981
*Fats*	34.39 ± 6.81	34.36 ± 5.49	0.996
**Fibers (%/kcal)**	3.55 ± 0.54	3.65 ± 0.50	0.697
**Amino Acids (%/total protein)**			
*Cystine*	2.05 ± 0.55	1.53 ± 0.43	**0.035**
*Phenylalanine*	7.27 ± 2.47	5.34 ± 1.21	**0.040**
*Tyrosine*	5.59 ± 2.06	3.93 ± 0.96	**0.034**
*Valine*	8.33 ± 2.91	5.87 ± 1.31	**0.025**
*Arginine*	8.52 ± 2.62	6.17 ± 1.50	**0.025**
*Histidine*	4.13 ± 1.44	2.97 ± 0.74	**0.035**
*Alanine*	7.22 ± 2.36	5.30 ± 1.29	**0.037**
*Aspartic acid*	14.32 ± 4.03	10.59 ± 2.27	**0.020**
*Glycine*	5.78 ± 1.67	4.34 ± 1.13	**0.039**
*Proline*	10.63 ± 4.48	7.43 ± 2.08	**0.055**
*Serine*	7.41 ± 2.56	5.28 ± 1.19	**0.028**

The table presents the values as mean ± standard deviation (mean ± SD). *p* represents the significance level of the differences between the normal group and the short group, as determined by independent-samples *t*-test. kcal = kilocalories; INSD = insomnia of normal sleep duration; ISSD = insomnia of short sleep duration. ***p*** < 0.05 are shown in **bold**.

**Table 3 biomolecules-14-00419-t003:** Bacterial metabolic pathway-associated metabolites significantly upregulated in ISSD participants.

Metabolite Name	Formula	RT (min)	Log_2_ Fold Change	*p*-Value
Benzophenone	C13H10O	16.841	3.23	0.001
Pyrogallol	C6H6O3	1.765	0.63	0.009
5-aminopental	C5H11NO	2.034	0.32	0.012
Butyl acrylate	C7H12O2	8.525	0.45	0.015
Kojic acid	C6H6O4	1.776	0.47	0.027
Deoxycholic acid	C24H40O4	19.577	0.84	0.031
trans-anethole	C10H12O	19.625	0.87	0.040
5-carboxyvanillic acid	C9H8O6	4.575	1.55	0.049

*p* represents the significance level of the differences between the normal and short groups, as determined using the *t*-test. RT—retention time; Log_2_ fold change—the average relative change on a Log_2_ scale. The Log_2_ fold change column represents the relative difference between the groups on a Log_2_ scale, where a positive value indicates higher levels in the ISSD group compared to the ISND group.

## Data Availability

Data are contained within the article and Appendix A.

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
