# Peer review of "Microbiota Metabolite Profiles and Dietary Intake in Older Individuals with Insomnia of Short vs. Normal Sleep Duration"

_biomolecules, 2024, doi:10.3390/biom14040419_

Round 1

Reviewer 1 Report

Comments and Suggestions for Authors

1. p6L214-216 A distinct metabolite profile was obtained for the two groups (Figure 1), with 82 metabolites demonstrating significantly higher levels in short sleep group.

In the volcano plot, the ISSD group is plotted to the left of the Log FC, which appears to indicate that all metabolites are reduced in the ISSD group compared to the INSD group. Also, isn't it wrong that the INSD is plotted since this volcano plot shows the variation of the ISSD versus the INSD?

2. In Table 3, eight of 82 metabolites have -0.32~-3.23 logFC, but they are higher levels in the ISSD group than the INSD group according to the sentence p6L214-216. This point and a sentence "P8L256 DCA, a gut metabolite associated with dysbiosis and disease, was upregulte in ISSD older adults" seem to contradict the results.

Author Response

We would like to thank the reviewer for his/her thoughtful and helpful comments, which have significantly improved our manuscript. We have addressed each comment herein in bold and, where applicable, marked the changes within the manuscript. We have included page and line numbers in our responses below for ease of reference.

  1. p6L214-216 A distinct metabolite profile was obtained for the two groups (Figure 1), with 82 metabolites demonstrating significantly higher levels in short sleep group. In the volcano plot, the ISSD group is plotted to the left of the Log FC, which appears to indicate that all metabolites are reduced in the ISSD group compared to the INSD group. Also, isn't it wrong that the INSD is plotted since this volcano plot shows the variation of the ISSD versus the INSD?

Thank you for the important comment. We corrected the volcano plot and its description. The revised Figure 1 shows the downregulated/upregulated metabolites in the ISSD group p8 lines 229-238. Furthermore, the explanation regarding the metabolomic analysis was corrected accordingly p7, lines 222-226.

  1. In Table 3, eight of 82 metabolites have -0.32~-3.23 logFC, but they are higher levels in the ISSD group than the INSD group according to the sentence p6L214-216. This point and a sentence "P8L256 DCA, a gut metabolite associated with dysbiosis and disease, was upregulate in ISSD older adults" seem to contradict the results.

Considering the correction done in the previous question, the results in Table 3 were revised and now show the bacterial metabolic pathway-associated metabolites significantly upregulated in ISSD participants.

Reviewer 2 Report

Comments and Suggestions for Authors

Manuscript number: biomolecules-2901092

Title: Microbiota Metabolite Profiles and Dietary Intake in Older Individuals with Insomnia of Short vs. Normal Sleep Duration

Summary:

The manuscript focuses on a critical evaluation of the dietary patterns and microbiota metabolites in older adults experiencing insomnia with short versus normal sleep duration (ISSD and INSD, respectively). This study brings to light the significant roles that diet and microbial metabolites play in insomnia, particularly concerning sleep duration. It emphasizes the potential of leveraging targeted dietary interventions and gut microbiota modulation as novel therapeutic approaches for managing insomnia, thus presenting a novel research idea that contributes to the development of understanding and treatment directions for sleep disorders in the elderly. This research is novel and presents some interesting explorations.

Recommendation:

The authors introduce a novel research idea as it enhances the understanding and development of treatment approaches for sleep disorders in older adults through the exploration of sleep, diet, and its interaction with gut microbiota. The manuscript presents detailed results on the variability of fecal metabolites in older adults with sleep disorders but lacks strong research results to support the influence of diet on sleep disorders in this demographic. It would be beneficial if the authors could provide additional information on the correlation between fecal metabolites and diet in influencing sleep disorders in the elderly. Based on these considerations, the study merits publication following minor revisions.

General Comments:

1. The impact of diet on sleep disorders in older adults is broadly defined, focusing only on the comparison of Total Energy (kcal/day), Macronutrients (%/kcal), Fibers (%/kcal), and Amino Acids (%/total protein) between groups, which appears insufficient. It is recommended to refine the definition and explore the data more precisely. For example, consider detailing the differences in various components that may explain the distinct fecal metabolite profiles observed between the two groups.

2. In section 2.3, Food frequency questionnaire (Data collection), please clarify how the Food Frequency Questionnaire (FFQ) was utilized to calculate Amino Acids (%/total protein). Specifically, describe the assays (e.g., UHPLC) employed to verify the presence of Amino Acids (%/total protein) in the samples, if possible. This clarification will enhance the manuscript's methodological transparency and allow readers to better understand the process involved in deriving key dietary data.

Author Response

We would like to thank the reviewer for his/her thoughtful and helpful comments, which have significantly improved our manuscript. We have addressed each comment herein in bold and, where applicable, marked the changes within the manuscript. We have included page and line numbers in our responses below for ease of reference.

Recommendation:

The authors introduce a novel research idea as it enhances the understanding and development of treatment approaches for sleep disorders in older adults through the exploration of sleep, diet, and its interaction with gut microbiota. The manuscript presents detailed results on the variability of fecal metabolites in older adults with sleep disorders but lacks strong research results to support the influence of diet on sleep disorders in this demographic. It would be beneficial if the authors could provide additional information on the correlation between fecal metabolites and diet in influencing sleep disorders in the elderly. Based on these considerations, the study merits publication following minor revisions.

We appreciate the constructive feedback and acknowledge the importance of elucidating the correlation between fecal metabolites, diet, and their influence on sleep disorders in the elderly. However, metabolomics studies addressing the interaction of these three factors are rare/nonexistent. To address the reviewers’ suggestion, we present additional information based on recent findings (the text was added to the “Introduction” section of the revised manuscript p2, lines 86-91).

General Comments:

  1. The impact of diet on sleep disorders in older adults is broadly defined, focusing only on the comparison of Total Energy (kcal/day), Macronutrients (%/kcal), Fibers (%/kcal), and Amino Acids (%/total protein) between groups, which appears insufficient. It is recommended to refine the definition and explore the data more precisely. For example, consider detailing the differences in various components that may explain the distinct fecal metabolite profiles observed between the two groups.

We appreciate your comment. We performed the statistical analysis in Table 2 on all the data extracted from the FFQ questionnaire. Since ISSD had a significant lower energy intake in comparison with the ISND group, lower levels of all macro and micronutrients were expected to be lower in the ISSD group, creating bias. Therefore, we calculated the % of each dietary compound within its category (e.g., % amino acids/ total protein), and performed the statistical analysis between both groups. No significant differences in fatty acids, minerals, and vitamins percentages, were detected between the groups. The information was added to p7, lines 204-212. All the raw data and the statistical analysis is shown in Supplementary data S1.

Moreover, we would like to clarify that correlations between diet composition and metabolomics are impossible in this study since we performed an untargeted analysis of the samples. 

  1. In section 2.3, Food frequency questionnaire (Data collection), please clarify how the Food Frequency Questionnaire (FFQ) was utilized to calculate Amino Acids (%/total protein). Specifically, describe the assays (e.g., UHPLC) employed to verify the presence of Amino Acids (%/total protein) in the samples, if possible. This clarification will enhance the manuscript's methodological transparency and allow readers to better understand the process involved in deriving key dietary data.

The calculation was as follows: We divided each amino acid by the total protein amount, resulting in the relative fraction of each amino acid (% amino acid/total protein). We would like to reinforce the fact that the reason for calculating the percentage of amino acid / total protein was because of the significant differences between the total calories consumed in each group. In addition, we would like to clarify that we did not use UHPLC to determine the amino acid content.

Round 2

Reviewer 1 Report

Comments and Suggestions for Authors

I believe this has been appropriately revised in response to my comments. As an additional comment, L263-264, "The orange and blue regions represent metabolites that were significantly down-regulated and up-regulated in the ISSD group, respectively" should be corrected to state that the orange region represents upregulated and the blue region represents downregulated.

Author Response

We thank the reviewer for his/her comment. The text was corrected accordingly.